# Reduced Expression of *PRX2*/*ATPRX1*, *PRX8*, *PRX35*, and *PRX73* Affects Cell Elongation, Vegetative Growth, and Vasculature Structures in *Arabidopsis thaliana*

**DOI:** 10.3390/plants11233353

**Published:** 2022-12-02

**Authors:** Yu Jeong Jeong, Young-Cheon Kim, June Seung Lee, Dong-Gwan Kim, Jeong Hwan Lee

**Affiliations:** 1School of Biological Sciences, Seoul National University, Seoul 08826, Republic of Korea; 2Department of Chemistry, College of Natural Sciences, Seoul National University, Seoul 08826, Republic of Korea; 3Division of Life Sciences, Jeonbuk National University, Jeonju 54896, Republic of Korea; 4Department of Life Science, Chung-Ang University, Seoul 06974, Republic of Korea; 5Department of Bioindustry and Bioresource Engineering, Sejong University, Seoul 05006, Republic of Korea

**Keywords:** cell elongation, class III peroxidase, *PRX2*/*ATPRX1*, *PRX8*, *PRX35*, *PRX73*, vasculature structure, vegetative growth

## Abstract

Class III peroxidases (PRXs) are involved in a broad spectrum of physiological and developmental processes throughout the life cycle of plants. However, the specific function of each PRX member in the family remains largely unknown. In this study, we selected four class III peroxidase genes (*PRX2*/*ATPRX1*, *PRX8*, *PRX35*, and *PRX73*) from a previous genome-wide transcriptome analysis, and performed phenotypic and morphological analyses, including histochemical staining, in *PRX2RNAi*, *PRX8RNAi*, *PRX35RNAi*, and *PRX73RNAi* plants. The reduced mRNA levels of corresponding PRX genes in *PRX2RNAi*, *PRX8RNAi*, *PRX35RNAi*, and *PRX73RNAi* seedlings resulted in elongated hypocotyls and roots, and slightly faster vegetative growth. To investigate internal structural changes in the vasculature, we performed histochemical staining, which revealed alterations in cell wall structures in the main vasculature of hypocotyls, stems, and roots of each *PRXRNAi* plant compared to wild-type (Col-0) plants. Furthermore, we found that *PRX35RNAi* plants displayed the decrease in the cell wall in vascular regions, which are involved in downregulation of lignin biosynthesis and biosynthesis-regulated genes’ expression. Taken together, these results indicated that the reduced expression levels of *PRX2*/*ATPRX1*, *PRX8*, *PRX35*, and *PRX73* affected hypocotyl and root elongation, vegetative growth, and the vasculature structures in hypocotyl, stem, and root tissues, suggesting that the four class III *PRX* genes play roles in plant developmental processes.

## 1. Introduction

Peroxidases (EC 1.11.1; PRXs), which catalyze oxidative reactions involving hydrogen peroxide (H_2_O_2_) as an electron acceptor, are categorized into at least four types in higher plants: glutathione peroxidase, catalase, ascorbate peroxidase (class I peroxidase), and classical plant peroxidases (class III peroxidases) [1]. Among them, class ΙΙΙ peroxidases (EC 1.11.1.7) are secretory enzymes and plant-specific heme oxidoreductases encoded by a large multigene family in late divergent plants [2]. In *Arabidopsis thaliana*, PRXs encoded by 73 genes are present as large multigene families, and the corresponding genes are expressed in a variety of tissues and developmental stages, albeit the substrate specificities of different PRX enzymes may be similar [3,4,5]. Thus, PRXs are involved in a diverse range of physiological and developmental processes during plant growth and development [6,7,8]. For instance, PRXs function in processes that occur from germination to aging, including cell elongation [9], cell wall metabolism [10,11], lignification [12], suberization [13], the catabolism of auxin and anthocyanin [14,15,16], secondary metabolism [17], and oxidative and biotic stresses [18,19]. These enzymes are also components of environmental stress-induced antioxidant systems [20]. These findings suggest that individual class III *PRX* genes have very specific functions in plant growth and development, or under abiotic and biotic stress conditions.

Lignification, a classic function of PRXs, is a cell wall-strengthening process that occurs in xylem tissues during tissue differentiation as well as during normal growth and defense responses [1]. Lignin, a major component of plant cell walls that is essential for plant stiffness, is mainly deposited in the vascular tissue. This compound provides structural strength to the cell wall, allowing the efficient water transport of xylem vessel elements and support of the fiber cells during vertical growth. Lignin is an aromatic biopolymer, and its metabolism contributes to plant growth, tissue formation, organ development, environmental stress protection, and the inhibition of pathogen invasion. Class III PRXs have been identified in functional analyses of *PRXs* based on the phenotype, expression, and biochemical characterization of mutants and transgenic plants as well as in studies of lignin-related PRXs [21,22,23,24,25]. Several *PRXs* identified from reverse genetic studies in *Arabidopsis*, including *PRX2*/*ATPRX1*, *PRX3*, *PRX4*, *PRX25*, *PRX52*, *PRX64*, *PRX71*, and *PRX72*, participate in various aspects of lignification such as lignin accumulation and composition [1]. For instance, T-DNA insertional *prx2*, *prx25*, or *prx71* single mutants in *Arabidopsis* showed both a decreased total lignin content and altered lignin structure, or only an altered lignin structure, whereas the three double mutants exhibited an altered lignin content and structure [6]. T-DNA mutations in *PRX4*, *PRX52*, and *PRX72* also led to a reduced lignin content and altered lignin composition [26,27]. Furthermore, artificial microRNA knockdown lines of *PRX64*, driven by an endodermis-specific promoter, affected the formation of Casparian strips as a lignin-based paracellular diffusion barrier in root cells [28]. These results suggest that many PRXs play an important role in lignification and that each PRX may be involved in the lignification process in specific organs and cell types or at specific growth and developmental stages.

Lignin is not easily hydrolyzed in lignin-saturated cell walls, but delignification also has a high chemical cost which negatively affects the production of biofuel [29]. Therefore, the biotechnology used to reduce lignin content and modify the lignin structure has received much attention; however, changes in lignin biosynthesis in the xylem often result in growth defects. Nevertheless, the development of technologies to solve this problem is expected to reduce the recalcitrance of biomass caused by lignin and increase the yields of extractable cellulose and fermentable sugar for pulp or biofuel production. In recent years, biotechnological approaches to modulating the lignin content have been used to optimize the plant biomass in various industries [30,31]. The study of lignin biosynthesis and its function has therefore had significant impacts on industry, agriculture, and other human activities.

Because brassinosteroids (BRs), as plant growth-promoting steroid hormones, are involved in *phytochrome B* (*phyB*)-mediated hypocotyl growth [32,33], we previously performed a genome-wide transcriptome analysis in *phyB-77* and *brassinosteroid insensitive 1-5* (*bri1-5*) mutants, and found that the expression levels of *PRX2*/*ATPRX1* and *PRX73* were differentially regulated between *phyB-77* and *bri1-5* mutants, and both *PRX2*/*ATPRX1* and *PRX73* RNAi transgenic plants showed elongated hypocotyls [34]. Because some PRXs are reported to play a role in plant developmental processes, such as hypocotyl and root growth [9,35,36,37], we aimed to assess the involvement of four *PRX* genes (*PRX2*/*ATPRX1*, *PRX8*, *PRX35*, and *PRX73*) in various developmental processes. Therefore, we performed a variety of phenotypical, histochemical, and molecular analyses using each knockdown plant for *PRX2*/*ATPRX1*, *PRX8*, *PRX35*, and *PRX73* genes using RNA interference (RNAi) technology.

## 2. Results

### 2.1. Identification of Four PRX Genes Belonging to Class III Peroxidase

In our previous research, we identified 624 genes co-regulated by BRs and light through a large-scale transcriptome analysis of *bri1-5*, *phyB*, and *bri1-5*/*phyB* mutants (GEO; http://www.ncbi.nlm.nih.gov/projects/geo/ (accessed on 30 November 2022); accession number GSE46456) [34]. In this study, we carried out a hierarchical cluster heat map analysis of co-regulated genes that were differentially expressed in *bri1-5* and *phyB* mutants. We found that after filtration with a modified criterion, (log2 fold change ([FC]) ≥ |1| and adjusted *p* < 0.01), four *PRX* genes (*PRX2*/*ATPRX1*, *PRX8*, *PRX35*, and *PRX73*) were up-regulated in *bri1-5*, but down-regulated in *phyB* (Appendix A). The phylogenetic tree analysis also showed that *PRX2/ATPRX1* was similarly clustered with *PRX8*, whereas *PRX35* and *PRX73* were clustered within the same group (Appendix A). Furthermore, the comparison of nucleotide sequences among the four *PRX* genes revealed 75–80% identity rates (Appendix A). We then chose four *PRX* genes (*PRX2*/*ATPRX1*, *PRX8*, *PRX35*, and *PRX73*) to further investigate their biological functions.

### 2.2. The Effect of RNAi-Mediated Silencing on PRX2/ATPRX1, PRX8, PRX35, and PRX73 Expression

Because PRX genes have known to affect several plant developmental processes [9,35,36,37], we firstly obtained an available T-DNA insertional mutants for each PRX gene (SALK_085028 (*PRX2*/*ATPRX1*), SALK_130310 (*PRX8*), SALK_119795 (*PRX35*), and SALK_010873 (*PRX73*)) from the Arabidopsis Biological Resource Center (ABRC) and confirmed each mutant allele carrying a T-DNA insertion via DNA-PCR genotyping (Appendix A). However, we found that the introduction of a single T-DNA insertion in *prx2/atprx1*, *prx8*, or *prx73* mutants, in which T-DNA insertions were found in the 3′-untranslated region (UTR), 1st exon, or promoter, respectively, caused no apparent phenotypic changes (Appendix A). In the *prx35* mutants, in which a T-DNA was inserted in the promoter, they showed a slightly early flowering and more accelerated leaf senescence phenotypes, compared with the wild-type (Col-0) plants (Appendix A). In order to generate each RNAi plant for four *PRX* genes, we first compared the identities of nucleotide sequences between four *PRX* genes and *PRX44*, with similar nucleotide sequences against the four *PRX* genes (Appendix A). Because the nucleotide sequences within coding regions showed high identity rates (75–80%) and it was difficult to select each specific region of the four *PRX* genes for RNA silencing, we decided to use each full-length open reading frame (ORF) region of the four *PRX* genes (Appendix A). We generated four RNAi constructs for *PRX2*/*ATPRX1*, *PRX8*, *PRX35*, and *PRX73* (Figure 1a) and transformed them into wild-type *Arabidopsis* (Col-0) plants. We designated them *PRX2RNAi*, *PRX8RNAi*, *PRX35RNAi*, and *PRX73RNAi* for *PRX2*/*ATPRX1*, *PRX8*, *PRX35*, and *PRX73*, respectively. After homozygous T_3_ selection, knockdown individuals were selected for each line, and the two independent lines with the lowest expression levels for each RNAi construct were selected (Figure 1b). For instance, the mRNA levels of each target gene were significantly reduced in the *PRX2RNAi* (#4-7 and #8-2), *PRX8RNAi* (#1-6 and #13-3), *PRX35RNAi* (#12-6 and #19-8), and *PRX73RNAi* (#2-6 and #10-1) plants by 39–96%, compared to the wild-type (Col-0) plants. This result indicated that a single *PRX2RNAi*, *PRX8RNAi*, *PRX35RNAi*, or *PRX73RNAi* construct can suppress the corresponding *PRX* gene targets.

To investigate the effect of the RNA silencing on the suppression of other target *PRX* genes in *PRX2RNAi*, *PRX8RNAi*, *PRX35RNAi*, and *PRX73RNAi* plants, we performed a RT–qPCR analysis of the four respective homozygous RNAi lines using *PRX2*/*ATPRX1*, *PRX8*, *PRX35*, and *PRX73* gene-specific primers (Appendix A). We found that the expression levels of the other target *PRX* genes were slightly reduced or remained unaltered in *PRX8RNAi*, *PRX35RNAi*, and *PRX73RNAi* plants, albeit the *PRX8*, *PRX35*, or *PRX73* mRNA levels were 30–50% downregulated in *PRX2RNAi* plants (Figure 1c). This result indicated some differences in the effectiveness of the respective RNAi constructs in silencing their target genes vs. the other three *PRX* genes [7,8,35,36,37].

### 2.3. Growth and Developmental Changes in Four PRXsRNAi Plants

Because some *PRX* genes are involved in a diverse range of developmental processes during plant growth and development [7,8], we observed morphological changes at the early developmental stage in the *PRX2RNAi*, *PRX8RNAi*, *PRX35RNAi*, and *PRX73RNAi* plants. Interestingly, the resulting four homozygous RNAi knockdown plants grown for six days under long-day (LD) conditions showed longer hypocotyls and roots than the wild-type (Col-0) plants (Figure 2a,b). These altered hypocotyl and root phenotypes observed in the four RNAi knockdown plants were also observed in other *prx* mutants [35,36,37]. Furthermore, the 2-week-old *PRX2RNAi*, *PRX8RNAi*, *PRX35RNAi*, and *PRX73RNAi* plants grown in soil had morphologically larger rosette leaves and grew slightly faster than the wild-type (Col-0) plants (Figure 2c). However, the effects of the other *PRX* genes’ suppression, observed in *PRX2RNAi*, *PRX8RNAi*, and *PRX73RNAi* plants, did not enhance the changes in hypocotyl and root elongation or rosette leaf size (Figure 1c and Figure 2). Our results indicated that the RNAi-mediated silencing of *PRX2*/*ATPRX1*, *PRX8*, *PRX35*, and *PRX73* affected the hypocotyl or root and leaf growth, suggesting the involvement of these four *PRX* genes at an early developmental stage.

We observed that *PRX2RNAi*, *PRX8RNAi*, *PRX35RNAi*, and *PRX73RNAi* plants grew faster than the wild-type (Col-0) plants at an early developmental stage (Figure 2c). To characterize the morphological changes between the wild-type (Col-0) and the four *RNAi* plants in more detail, we compared and analyzed plant growth phenotypes such as the leaf petiole length, blade length, and leaf width under LD conditions. We found that the 30-day-old soil-grown *PRX2RNAi*, *PRX8RNAi*, *PRX35RNAi*, and *PRX73RNAi* plants grew slightly faster than the wild-type (Col-0) plants (Figure 3a). The morphological parameters (petiole length, blade length, and blade width) of the rosette leaves in the *PRX2RNAi* and *PRX73RNAi* plants significantly altered compared with those of the wild-type (Col-0) plants, whereas leaves of some *PRX8RNAi* and *PRX35RNAi* lines showed reduced or no significant changes in their petiole length and leaf width (Figure 3b,c).

We also examined the bolting days, whose start was defined as the point at which the first internode stem length was 1 cm [38], in four *RNAi* plants grown under LD conditions. Their bolting occurred approximately five days earlier in *PRX2RNAi*, *PRX8RNAi*, *PRX35RNAi*, and *PRX73RNAi* plants than in the wild-type (Col-0) plants (Figure 3d), indicating that the four *RNAi* plants had slightly early-flowering phenotypes. After bolting, changes in the reproductive organs such as the inflorescences and siliques were observed in the four *RNAi* plants. For instance, the *PRX35RNAi* plants had smaller inflorescences and shorter siliques (Appendix A). These results indicated that the RNA silencing of *PRX2*/*ATPRX1*, *PRX8*, *PRX35*, and *PRX73* caused morphological changes in vegetative and reproductive organs.

Because some *RNAi* plants showed more accelerated leaf senescence phenotypes (Figure 3b), we measured the chlorophylls and anthocyanin contents in the four *RNAi* plants. Among the 45-day-old plants grown in soil, rosette leaves of *PRX2RNAi*, *PRX8RNAi*, and *PRX73RNAi* plants, but not those of a *PRX35RNAi* line (#12-6), showed decreased chlorophyll contents and increased anthocyanin contents compared with the wild-type (Col-0) plants (Figure 3e,f). These results indicated that the RNAi-mediated silencing of *PRX2*/*ATPRX1*, *PRX8*, *PRX35*, and *PRX73* caused a variety of morphological changes at late developmental stages.

### 2.4. Phenotypic Changes of Vascular Patterns in Stems of Four PRXRNAi Plants

Several studies have revealed that defects in some *PRX* genes affect vascular development [1,6,26,27]. To investigate the differences in primary vascular patterns induced by the RNA silencing of each *PRX* gene, we analyzed the basal section of the main stems of 6-week-old wild-type (Col-0) and four respective *PRXRNAi* plants via histochemical staining with toluidine blue. The toluidine blue staining revealed significant changes in the stained cells between the wild-type (Col-0) and four *PRXRNAi* plants (Figure 4a,b and Appendix A). Reduced numbers of cells were observed in the xylem layers and interfascicular fibers of four *PRXRNAi* lines, compared with the wild-type (Col-0) plants (Figure 4c). Thus, the average cell numbers of xylem and interfascicular fiber cells along the vascular ring were reduced in the *PRX2RNAi*, *PRX8RNAi*, *PRX35RNAi*, and *PRX73RNAi* plants compared to the wild-type (Col-0) plants. Especially the *PRX35RNAi* plants, with the smallest stem diameter, also had a reduced number of vascular bundles (VBs) (Figure 4a,c and Appendix A). Furthermore, the cell size in the xylem and interfascicular fibers of some *PRXRNAi* lines was much larger than that in the wild-type (Col-0) plants (Figure 4b). These changes in vascular patterns were similar to those observed in three double mutants of *PRX2*/*ATPRX1*, *PRX25*, and *PRX71* [6]. Based on our results, the RNAi-mediated silencing of *PRX2*/*ATPRX1*, *PRX8*, *PRX35*, and *PRX73* partially led to defects in stem vascular patterns.

### 2.5. Phenotypic Changes in Hypocotyl and Root Vasculature Structures in PRX35RNAi Plants

Because two independent *PRX35RNAi* lines (#12-6 and #19-8), in which the expression levels of the other three *PRXs* almost remained unaltered (Figure 1c), showed severe changes in the size of the main stem and the number of vascular bundles (Figure 4 and Appendix A), we microscopically examined cross-sections of the vascular tissues of hypocotyls and roots between wild-type (Col-0) and *PRX35RNAi* plants (#12-6 and #19-8) grown for 6 days. Our microscopic observations of the roots and hypocotyls of the *PRX35RNAi* plants revealed that the overall anatomical features of their vascular bundles were altered, compared with the wild-type (Col-0) plants, and that the procambium regions (Pc; red square) were also slightly reduced (Figure 5a and Appendix A). The arrangement of the xylem, cambium, and phloem in the cross-sections of the upper portions of the hypocotyls of two independent *PRX35RNAi* plants was not significantly different from that in the wild-type (Col-0) plants (Figure 5a and Appendix A), but the numbers of xylem and procambial cells (white circle and red square, respectively) were reduced. In addition, we observed that the xylem cells (white circle) in the hypocotyls of two independent *PRX35RNAi* plants were larger in diameter than those in the wild-type (Col-0) plants (Figure 5a and Appendix A). This result indicated that the number and size of vascular cells were altered in the hypocotyls of *PRX35RNAi* plants, albeit the basal vascular structures were retained.

We also investigated the ultrastructure of the cell walls in transverse sections of roots between wild-type (Col-0) and *PRX35RNAi* plants (#12-6) grown for 6 days. The transverse sections of the primary roots in the *PRX35RNAi* and wild-type (Col-0) seedlings consisted of a monolayer of epidermal, cortical, endodermal, and pericycle cells (Figure 5b) revealed substantial differences in the cell walls of each cell type in the roots of the *PRX35RNAi* plants (Figure 5c,d). Although the metaxylem (Mx) cells remained unaltered in the *PRX35RNAi* plants, the protoxylem (Px) cells were destroyed by the expansion of the surrounding cells. Moreover, the roots of the *PRX35RNAi* plants displayed an abnormal phloem pattern (CC, MSe, PPP, and PSe) (Figure 5d), and the cortex (Co) cell walls in the *PRX35RNAi* plants were thinner than those of the wild-type (Col-0) plants (Figure 5e). Based on our results, the RNAi-mediated silencing of *PRX35* caused defects in specific differentiated cell types in the hypocotyl and root vasculature structures.

### 2.6. Altered Vasculature Structures and Decreased mRNA Levels of Lignin Synthesis and Synthesis-Regulated Genes in Stems of PRX35RNAi Plants

To examine the histochemical analysis in the stems of 6-week-old wild-type (Col-0) and *PRX35RNAi* plants (#12-6), we stained the sections taken from the bases of the main stems with toluidine blue. The toluidine blue staining of the stem cross-sections revealed that the stained cell wall area in the xylem and interfascicular fibers decreased in the stems of the *PRX35RNAi* plants (Figure 6a). Although the shape and characteristics of the conduit were unchanged in *PRX35RNAi* plants (Figure 6a), the total number of xylem cells and interfascicular fibers decreased by more than 50%. The epidermal cell and vascular bundle numbers were 25% and 20% lower, respectively, in *PRX35RNAi* plants than in wild-type (Col-0) plants (Figure 6b). These results indicated that the cell wall area and cell numbers were significantly reduced in xylem cells and interfascicular fibers in stems of *PRX35RNAi* plants.

Lignin, as a component of the secondary cell walls, plays an important role in maintaining the plant structure and increasing the cell wall rigidity [39]. Because the cell wall area was reduced in the stems of 6-week-old *PRX35RNAi* plants (Figure 6a), we analyzed the mRNA expression levels of genes involved in the lignin synthesis and synthesis regulation in 6-week-old wild-type (Col-0) and *PRX35RNAi* plants (#12-6). We harvested the bottom, middle, and top stem parts, which correspond to non-elongating internodes, internodes near the cessation of elongation, and rapidly elongating internodes, respectively [40]. The expression levels of the genes encoding transferases or belonging to the common route of the phenylpropanoid-lignin biosynthesis pathway, such as phenylalanine ammonia lyase (PAL), cinnamic acid 4-hydroxylase (C4H), 4-coumarate CoA ligase 1 (4CL), hydroxycinnamoyl-CoA shikimate/quinate hydroxycinnamoyl transferase (HCT), coumaric acid 3-hydroxylase 1 (C3H), caffeoyl-CoA O-methyltransferase 1 (CCOAMT), ferulate 5-hydroxylase (F5H), and O-methyltransferase (COMT), as well as the gene encoding cinnamyl alcohol dehydrogenase 6 (CAD) acting as the final committed enzyme for biosynthesis of several lignin types, were down-regulated in all three stem regions of *PRX35RNAi* plants (Figure 6c, Appendix A). Among the numerous transcription factors involved in lignin biosynthesis regulation, *MYB58* and *MYB63* are well known as important regulators of lignin biosynthesis, which are specifically expressed in cells with thickened secondary walls [41,42,43]. The expression levels of both transcription factors were also reduced (Figure 6d, Appendix A). These results suggest that the alteration in vasculature structures observed in *PRX35RNAi* plants may result from a consequence of the accumulation of monolignols, such as *p*-hydroxyphenyl, guaiacyl, and syringyl units, caused by the decreased expression of lignin biosynthesis and biosynthesis-regulated genes.

## 3. Discussion

Class III PRXs perform diverse functions in plants, but only a few of these enzymes have been fully characterized. To better understand the roles of PRXs in plant growth and development, such as in cell elongation and vegetative growth, and in the vasculature in the hypocotyls, stems, and roots, we analyzed four class III *PRX* genes (*PRX2*/*ATPRX1*, *PRX8*, *PRX35*, and *PRX73*), selected via a genome-wide transcriptome analysis, and generated respective RNAi knockdown plants to characterize their functions. The phenotypic and morphological observations, including histochemical staining and gene expression analyses, revealed that the RNAi silencing of the four selected *PRX* genes resulted in elongated hypocotyls and roots, vegetative growth, and altered cell wall structures in the main vasculature in the hypocotyls, stems, and roots. An in-depth analysis of the *PRX35RNAi* plants further demonstrated that the RNAi-mediated specific silencing of *PRX35* led to a reduction in cell walls in the vascular regions and the downregulation of lignin biosynthesis and biosynthesis-regulated genes.

Several reports have shown that some *PRX* genes play roles in a variety of plant developmental processes [9,35,36,37]. In this study, we observed that the respective *RNAi* plants for each of the four *PRXs* showed elongated hypocotyl and root lengths (Figure 2a,b), suggesting that *PRX2*/*ATPRX1*, *PRX8*, *PRX35*, and *PRX73* may repress hypocotyl and root elongation. These morphological changes were also observed in *prx53* mutants [36]. Thus, it is likely that *PRX2*/*ATPRX1*, *PRX8*, *PRX35*, and *PRX73* may catalyze the cross-linking of cell wall compounds to repress cell elongation in the hypocotyls and roots [44,45,46,47,48,49]. However, considering that *PRX34* overexpressors and *prx33* mutants show longer and shorter roots, respectively [35], it is likely that *PRX* genes belonging to different phylogenetic clades may have opposite functions in cell elongation of the same tissues (Appendix A). One interesting phenotype shown in four *PRXRNAi* plants was faster vegetative growth, such as the increased size of rosette leaves at an early developmental stage (Figure 2c), which is partially consistent with a finding showing that *prx71* mutants display an enhanced rosette leaf size [9]. These results suggest that *PRX2*/*ATPRX1*, *PRX8*, *PRX35*, and *PRX73* may negatively regulate vegetative growth, such as for the leaf cell size at an early developmental stage. This speculation is supported by an observation that KUODA1 (KUA1) as a transcription factor specifically regulates cell expansion during leaf development by negatively modulating the expression levels of seven *PRX* genes, including *PRX8* and *PRX35* as characterized in this study [50]. Furthermore, several reports revealed that *prx72* and *prx17* mutants show the opposite effect on the flowering time [26,51], suggesting that a subset of *PRX* genes are involved in the control of the flowering time. Because the four *PRXRNAi* plants showed a slightly early flowering time (Figure 3a,d), it is worth noting that the four *PRX* genes investigated in this study may be negative regulators in the control of the flowering time. Our unexpected findings were that the four *PRXRNAi* plants showed accelerated leaf senescence phenotypes and had decreased chlorophyll contents with increased anthocyanin contents (Figure 3b,e,f). Because the peroxidase activities were increased and the superoxide radical was generated, thereby inducing the senescence pathway during senescence [52], our data were inconsistent with this notion. This suggests that the reduced expression levels of *PRX2*/*ATPRX1*, *PRX8*, *PRX35*, and *PRX73* observed in the respective *PRXRNAi* plants may lead to faster vegetative growth, thereby accelerating leaf senescence and changes in chlorophyll and anthocyanin contents. However, considering that the *35S* promoter-driven transgenes lead to ectopic or elevated expression in transgenic plants, we could not exclude the possibility that the respective *PRXRNAi* plants used in this study do not reflect the genuine function of the four class III *PRX* genes. Further studies on other knockout mutant alleles, or their own endogenous promoter-driven antisense and sense transgenic plants, would be informative for understanding the relevant functions such as the leaf cell size and changes in flowering time of the *PRX* genes.

Lignin is mainly deposited in the cell walls, causing secondary wall thickening in the xylem and interfascicular fibers during plant growth development [53]. Some of the lignin-polymerizing enzymes PRXs have been shown to affect stem lignification [54]. In this study, the RNAi silencing of the four *PRX* genes caused reduced cell numbers in the xylem layers and interfascicular fibers along the vascular ring in the stems (Figure 4 and Appendix A). This altered phenotype was consistent with previous findings showing that three double mutants among *prx2*, *prx25*, and *prx71* mutants and single *prx72* mutants showed reduced cell wall volume in interfascicular fibers, thereby causing reductions in cell wall thickness [6,26]. However, the changes in lignin monomer composition between the three double mutants (*prx2 prx25*, *prx2 prx71*, and *prx25 prx71* mutants) and single *prx72* mutants were different (syringyl/guaiacyl (*S*/*G*) ratio = increase and decrease, respectively), albeit the total lignin contents were reduced. Especially the *PRX35RNAi* plants showed some changes in specific differentiated cell types in the hypocotyl and root vasculature structures, but not changes in the basal vascular structures (Figure 5, Figure 6a,b and Appendix A), which is consistent with a finding showing that the significant change in the thickening of the primary cell walls is found in *prx52* mutants [27]. Considering the previous observations that other *PRX* genes differentially affect the lignin structure, content, and composition [6,26,27], our results suggest that *PRX2*/*ATPRX1*, *PRX8*, *PRX35*, and *PRX73* may affect the lignified levels in the vasculature. This possibility is further supported by our observation that the expression levels of lignin biosynthesis and synthesis-regulated genes were significantly down-regulated in the *PRX35RNAi* plants (Figure 6c,d), which is consistent with the data obtained from *prx52* mutants [27]. Considering the preferential expression patterns in the vasculature regions of a subset of lignin biosynthesis and biosynthesis-regulation genes, and their mutant phenotypes observed in vascular tissues of the leaves and stems [41,42,43], it is likely that a reduced expression level of *PRX35* in the *PRX35RNAi* plants may lead to feedback inhibition caused by the accumulation of monolignols such as *p*-hydroxyphenyl, guaiacyl, and syringyl units [52,55] in the vasculature regions of the stems, thereby decreasing the cell numbers and lignin levels in the vasculature. This notion is further supported by the observations that the levels of lignin composition are significantly reduced in some mutants of a subset of lignin biosynthesis and biosynthesis-regulation genes [42,56,57,58,59]. Considering that the major function of class III *PRX* genes is in lignin modifications, such as alterations of the lignin structure and composition, further experiments involving lignin quantification in the four *PRXRNAi* lines, T-DNA mutants, or multiple combinations between the *PRXRNAi* lines and T-DNA mutants are needed to elucidate whether the four class III *PRX* genes are involved in lignin formation in the vasculature.

## 4. Materials and Methods

### 4.1. Plant Materials and Growth Conditions

*Arabidopsis* ecotype Columbia-0 (Col-0) was used as the wild-type control, because this accession was the genetic background for the transgenic lines used in this study. In addition, we obtained *Arabidopsis* T-DNA insertional mutants of four *PRX* genes (*PRX2*/*ATPRX1*, *PRX8*, *PRX35*, and *PRX73*) from ABRC [60]. The *Arabidopsis* seeds were sterilized with 75% ethanol containing 0.05% Tween-20 for 10 min, washed once in 100% ethanol, and then placed onto half-strength Murashige and Skoog (MS; Duchefa Biochemie, Haarlem, The Netherlands) plates containing 0.8% phytoagar. After stratification at 4 °C for 3 d, the plants were grown in a growth chamber at 23 °C under long-day (LD) conditions (16 h light/8 h dark) at a light intensity of 120 μmol m^−2^ s^−1^.

For growth analysis, the following parameters were measured: petiole length, leaf blade length, and leaf width of the fifth leaves of 30-day-old soil-grown plants (*n* ≥ 53) and silique length and leaf size of 6-week-old soil-grown plants (*n* ≥ 10). All plants were photographed and measured using Image Tool software (version 3.0, UTHSCSA, San Antonio, TX, USA), as described by Jeong et al. [61]. All measurements were repeated at least three times. For morphometric analysis of hypocotyls and roots, 6-day-old seedlings (*n* = 20) grown in half-strength MS medium were photographed and measured using Image Tool software [61]. All measurements were repeated at least three times. For the bolting-time analysis, plants (*n* = 17) with inflorescence stems 1 cm long were considered flowering [38]. All measurements were repeated at least three times.

### 4.2. Generation of Transgenic Arabidopsis Plants

To generate RNAi silencing constructs of target genes *PRX2*/*ATPRX1*, *PRX8*, *PRX35*, and *PRX73*, we amplified the full-length open reading frames (ORFs) (*PRX2*/*ATPRX1*: AT1G05250, 982 bp; *PRX8*: AT1G34510, 982 bp; *PRX35*: AT3G49960, 991 bp; and *PRX73*: AT5G67400, 991 bp) by PCR using gene-specific primers. The primers used for vector construction are listed in Appendix A. The polymerase chain reaction (PCR) products were subcloned into a pENTR/SD/D-TOPO vector (Invitrogen, Carlsbad, CA, USA) using LR Clonase (Thermo Fisher Scientific, Waltham, MA, USA), confirmed by sequencing, and finally cloned into a pB7GWIWG2 (II) RNAi destination vector (Invitrogen) to induce target gene silencing according to the Gateway system described by Jeong [62]. The resulting constructs [*p35S::RNAi-PRX2* (*PRX2RNAi*), *p35S::RNAi-PRX8* (*PRX8RNAi*), *p35S::RNAi-PRX35* (*PRX35RNAi*), and *p35S::RNAi-PRX73* (*PRX73RNAi*)] were verified by DNA sequencing and introduced into *Arabidopsis* plants using *Agrobacterium tumefaciens* strain GV3101 by the floral dip method [63], thereby generating *PRX2RNAi*, *PRX8RNAi*, *PRX35RNAi*, and *PRX73RNAi* transgenic lines. Through 50mg L^−1^ Basta (DL-phosphinothricin; Duchefa Biochemie, Haarlem, The Netherlands) screening, more than 10 to 20 transgenic lines were selected for each RNAi line. After Mendelian segregation, we selected single-copy homozygous T_3_ generation lines for each RNAi line and further analyzed at least two independent transgenic lines for each RNAi construct.

### 4.3. RNA Isolation and Gene Expression Analysis

Total RNA (2 μg) was isolated from 6-day-old seedlings grown in half-strength MS medium using the TRIzol reagent (Sigma-Aldrich, Burlington, MA, USA). Subsequently, total RNA was treated with DNase (Promega, Madison, WI, USA) as described by Jeong et al. [64,65]. cDNA was synthesized using reverse transcriptase (Fermentas, Seoul, Republic of Korea). Real-time quantitative PCR (RT–qPCR) analysis was performed on an AB7300 Real-time PCR instrument (Thermo Fisher Scientific) using SYBR Green PCR Master Mix (Sigma-Aldrich). All gene-specific primers for PRX, lignin biosynthesis, and lignin biosynthesis-regulated genes are listed in Appendix A, respectively. The reactions were performed with three biological replicates, and the data were analyzed using one-way ANOVA, followed by Tukey’s post hoc test or Student’s *t*-test. For all RT–qPCR analyses, the relative transcript levels were normalized to the expression of ubiquitin 10 (*UBQ10*).

### 4.4. Measurements of Chlorophyll and Anthocyanin Contents

Chlorophyll a, chlorophyll b, and total chlorophyll contents of rosette leaves were measured as described by Lichtenthaler [66]. Leaf samples (0.012 g) of 45-day-old soil-grown plants were frozen in liquid nitrogen and homogenized, and chlorophyll was extracted using 80% acetone. The absorbance was then measured at 647, 652, and 664 nm using a Spectronic Genesys 5 spectrophotometer and used to calculate chlorophyll content. All measurements were repeated at least three times.

The anthocyanin content of the rosette leaves was measured as previously described [67]. Leaf samples (0.012 g) of 45-day-old soil-grown plants were frozen in liquid nitrogen and homogenized. After extraction of anthocyanins for 1 d at 4 °C in 1 mL of 1% hydrochloric acid in methanol, the absorbance was measured at 530 and 657 nm. Relative anthocyanin concentrations were calculated according to the formula [A_530_ − (0.25 × A_657_)]. The relative amount of anthocyanin was defined as the product of the relative anthocyanin concentration and the extraction solution volume. One anthocyanin unit equaled one absorbance unit [A_530_ − (0.25 × A_657_)] in 1 mL of the extraction solution. All measurements were repeated at least three times.

### 4.5. Histochemical Staining

Freshly harvested tissues, including the primary inflorescence stem (2 cm from the base), elongation zone of roots, and middle zone of hypocotyls, were fixed in FAA solution (10% formaldehyde, 5% acetic acid, and 50% ethanol) and left overnight at 23 °C. Dehydration was performed using a graded ethanol series (50, 70, 80, 90, and 100%) at 23 °C, for 1 h per step. After dehydration, a portion of each tissue sample was infiltrated for 2 h in a solution containing 50% ethanol and 50% Technovit 7100 (Heraeus-Kulzer, Wehrheim, Germany). The samples were then incubated overnight in a solution of 1% Hardener I in a 100% Technovit 7100 resin. The remaining tissue after dehydration was infiltrated for 2 h in a solution containing 50% ethanol and 50% Technovit 7100. Polymerization was performed by adding hardener II. The embedded samples were cut usng a diamond knife on an ultramicrotome (Ultracut S, Leica Microsystems, Wetzlar, Germany) to a thickness of 3–5 μm, mounted on glass slides, and stained with 0.1% toluidine blue. Cross-sections were analyzed using a fluorescence microscope (Olympus BX 41, Tokyo, Japan). All measurements were repeated at least three times.

### 4.6. Transmission Electron Microscopy (TEM)

Freshly harvested tissues, including the primary inflorescence stems, roots, and hypocotyls, were fixed in 2.5% glutaraldehyde at 4 °C, after rinsing three times in 0.05 M cacodylate buffer (pH 6.9) containing 1% osmium tetroxide for 3 h at 4 °C. The fixed tissues were dehydrated using a graded ethanol series (30, 50, 70, 80, 90, and 100%) for 15 min at each step. The dehydrated samples were embedded in Spurr’s resin series [68] and polymerized at 60 °C for 8 h. The embedded samples were cut with a diamond knife using an Ultracut S ultramicrotome. Sections were mounted directly on 150-mesh copper grids and stained with toluidine blue. The grids were observed using a cryo-TEM (Tecnai F20 Cryo, FEI, The Netherlands), installed at the bio-imaging core facility of the Republic of Korea Institute of Science and Technology (KIST). All measurements were repeated at least three times.

### 4.7. Quantitative Vascular Analysis

Vascular bundles and cells were counted from microscope images using ImageJ software (http://rsb.info.nih.gov/ij/ (accessed on 1 January 1997). The data in the quantitative vascular analysis experiments (*n* = 10) were subjected to statistical analysis using the Student’s *t*-test. To measure cell wall thickness in interfascicular fibers and xylem (xylary fibers and vessels), the primary inflorescence stems (2 cm from the base) were cut into 2 mm pieces for ultrathin sectioning and TEM.

## Figures and Tables

**Figure 1 plants-11-03353-f001:**
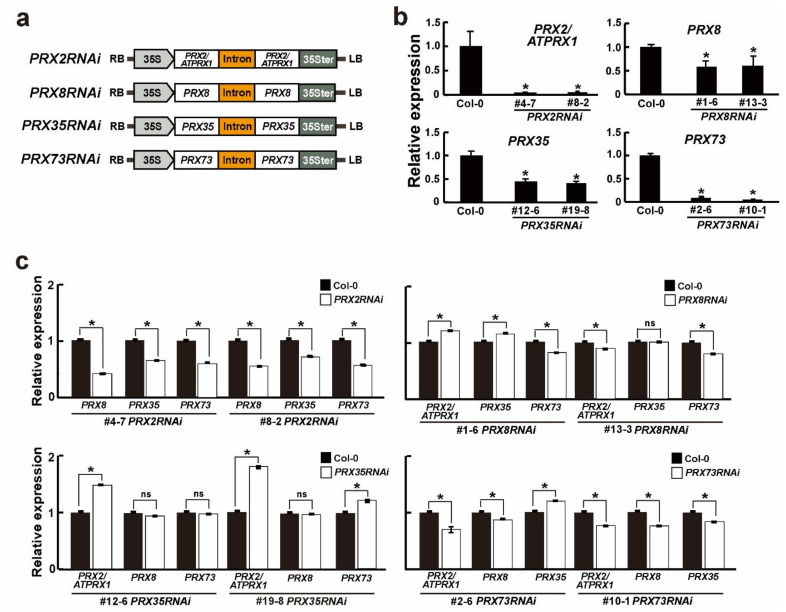
The expression of each *PRX* gene in the four *PRXRNAi* plants. (**a**) Schematic diagram of *PRX2RNAi*, *PRX8RNAi*, *PRX35RNAi*, and *PRX73RNAi* constructs. LB, RB, 35S, PRX, and 35Ster indicate the T-DNA left border, T-DNA right border, cauliflower mosaic virus (CaMV) *35S* promoter, each peroxidase ORF, and the CaMV *35S* terminator, respectively. (**b**,**c**) Relative expression levels of target or other *PRX* genes in wild-type (Col-0) and respective *PRXRNAi* plants grown for 6 days. Relative expression levels measured by RT–qPCR in wild-type (Col-0) plants were defined as 1.0. Error bars indicate the standard error of the mean. The *UBQ10* gene was used as an internal control. Asterisks indicate significant differences (*p* < 0.005, Student’s *t*-test). Note: ns–not significant.

**Figure 2 plants-11-03353-f002:**
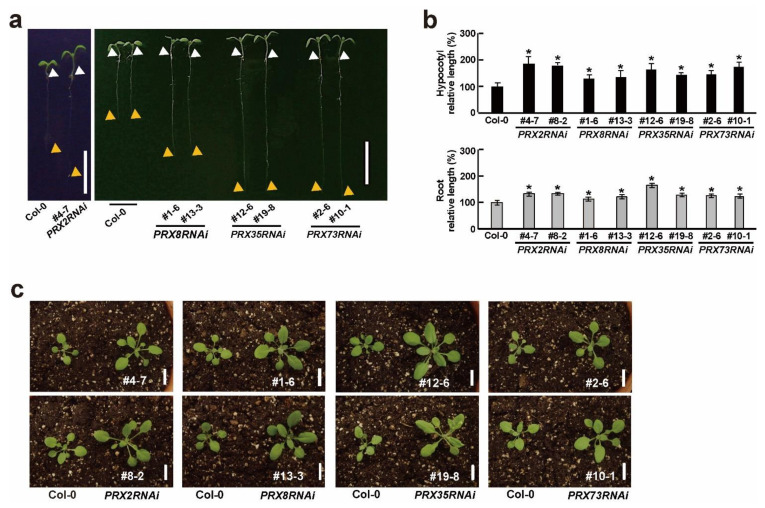
Morphological changes in the four *PRXRNAi* plants at early developmental stages. (**a**) Phenotypes of 6-day-old wild-type (Col-0) and respective *PRXRNAi* seedlings grown in half-strength MS medium. White and orange triangles indicate hypocotyl and root ends, respectively. Scale bar = 1 cm. (**b**) Relative lengths of hypocotyls and roots in 6-day-old grown wild-type (Col-0) and respective *PRXRNAi* seedlings. Error bars indicate standard error of the mean (*n* = 20). Significant differences (*p* < 0.001, Student’s *t*-test) are indicated by asterisks. (**c**) Phenotypes of 2-week-old soil-grown wild-type (Col-0) and respective *PRXRNAi* plants before bolting. Scale bar = 1 cm.

**Figure 3 plants-11-03353-f003:**
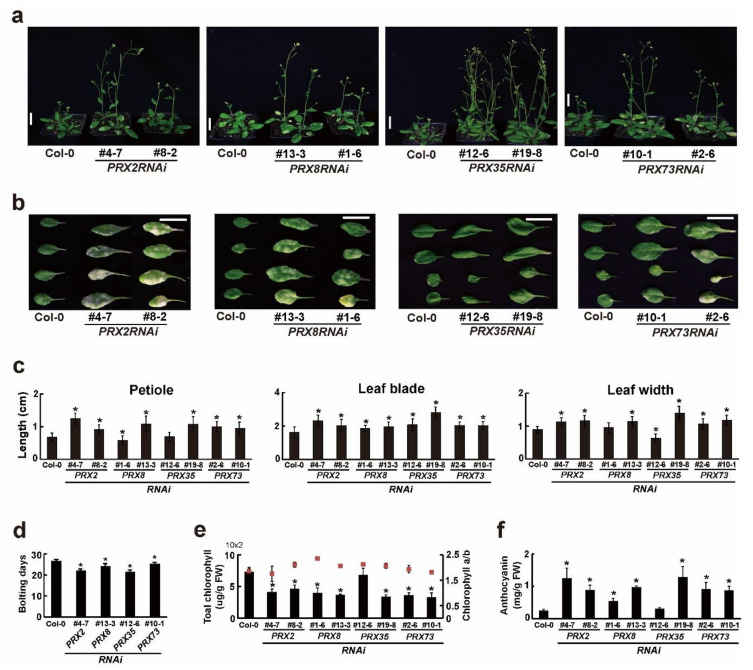
Morphological changes in four *PRXRNAi* plants at late developmental stages. (**a**) Phenotypes of 30-day-old soil-grown wild-type (Col-0) and respective *PRXRNAi* plants. Scale bar = 2 cm. (**b**) Phenotypes of third to sixth rosette leaves of wild-type (Col-0) and respective *PRXRNAi* plants grown in soil for 30 days. Scale bar = 1 cm. (**c**) Lengths of the leaf petiole, leaf blade, and leaf width in the fifth and sixth rosette leaves of 30-day-old soil-grown plants. Error bars indicate standard error of the mean (*n* ≥ 53). Asterisks indicate significant differences (*p* < 0.001, Student’s *t*-test). (**d**) Bolting days of wild-type (Col-0) and respective *PRXRNAi* plants (*n* = 17). Asterisks indicate significant differences (*p* < 0.001, Student’s *t*-test). (**e**,**f**) Total chlorophyll and (**e**) anthocyanin (**f**) contents in rosette leaves of 45-day-old soil-grown wild-type (Col-0) and respective PRXRNAi plants (*n* = 10). Red squares shown in (**e**) indicate the chlorophyll a/b contents. Asterisks indicate significant differences (*p* < 0.005, Student’s *t*-test).

**Figure 4 plants-11-03353-f004:**
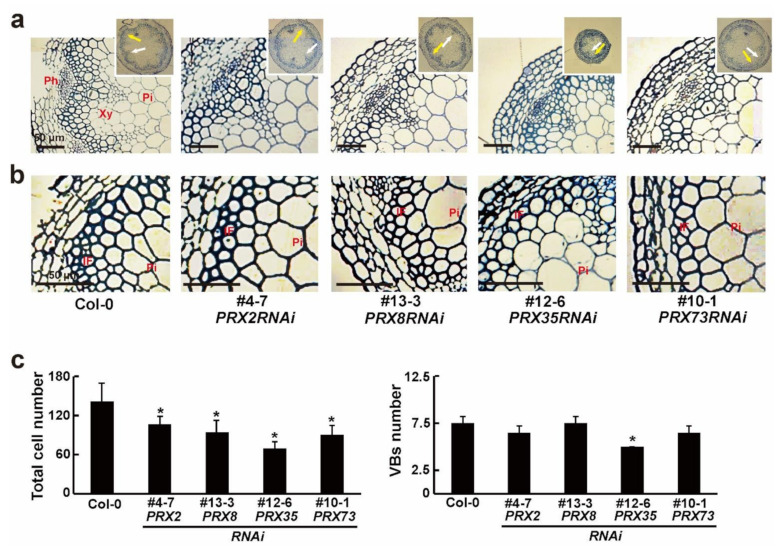
Transverse sections of basal main stems of wild-type (Col-0) and *PRXRNAi* plants. (**a**,**b**) Enlarged photos of the xylem (**a**) and interfascicular fiber (**b**) parts (upper right corner) in toluidine blue-stained sections of resin-embedded stems of soil-grown 6-week-old wild-type (Col-0) and respective *PRXRNAi* plants. White and yellow arrows indicate the xylem and interfascicular fibers, respectively. The phloem (Ph), xylem (Xy), pith (Pi), and interfascicular fiber (IF) parts are indicated. Scale bars = 50 μm. (**c**) Average total numbers of xylem cells and interfascicular fibers (**left**) along the vascular ring, and vascular bundles (**right**) in stems of soil-grown 6-week-old wild-type (Col-0) and respective *PRXRNAi* plants. Error bars indicate the standard error of the mean (*n* ≥ 10). Asterisks indicate significant differences (*p* < 0.005, Student’s *t*-test).

**Figure 5 plants-11-03353-f005:**
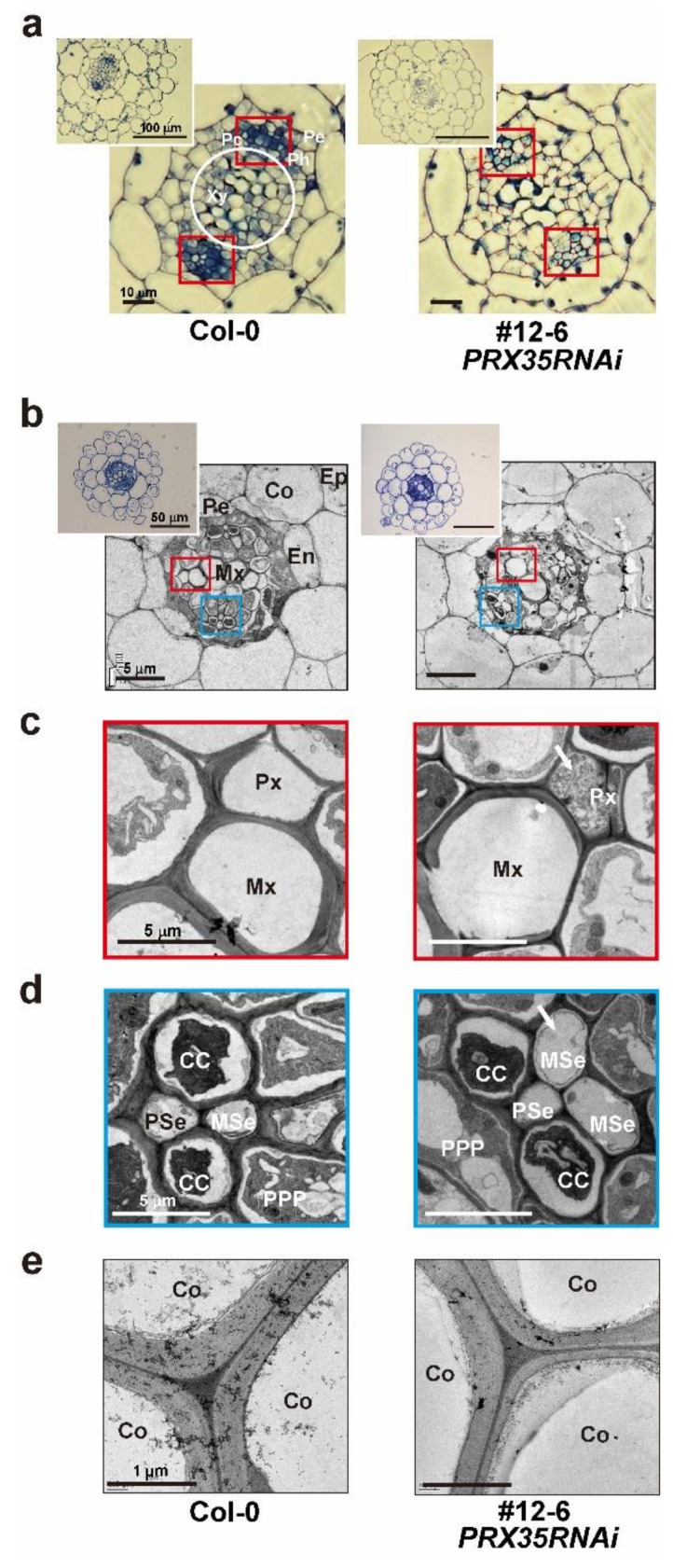
Transverse sections of hypocotyl and root regions in wild-type (Col-0) and *PRX35RNAi* seedlings. (**a**) Enlarged photos of vascular bundles (upper left corner) in toluidine blue-stained sections of resin-embedded upper hypocotyls in 6-day-old wild-type (Col-0) and *PRX35RNAi* seedlings. Scale bars are presented in photos. (**b**) Enlarged photos of vascular bundles (upper left corner) in toluidine blue-stained sections of resin-embedded upper hypocotyls in elongation zones of roots of whole 6-day-old seedlings. The red and blue boxes are magnified in (**c**,**d**). Scale bars are presented in photos. (**c**–**e**) Transmission electron microscopy (TEM) images of resin-embedded vascular bundles in the elongation zone of roots of the wild-type (Col-0) and *PRX35RNAi* seedlings shown in (**b**). Enlarged pictures of the red and blue boxes are shown in (**c)** and (**d**), respectively. White arrows indicate the abnormal part of the *PRX35RNAi* line. A comparison of the cell wall thicknesses of the cortexes between the wild-type (Col-0) and *PRX35RNAi* lines are represented in (**e**). The cortex (Co), endodermis (En), epidermis (Ep), metaphloem sieve element (MSe), metaxylem (Mx), procambium (Pc), pericycle (Pe), phloem (Ph), phloem companion cell (CC), protophloem pole pericycle cells (PPP), protophloem sieve element (PSe), protoxylem (Px), and xylem (Xy) cells are indicated. Scale bars are presented in photos.

**Figure 6 plants-11-03353-f006:**
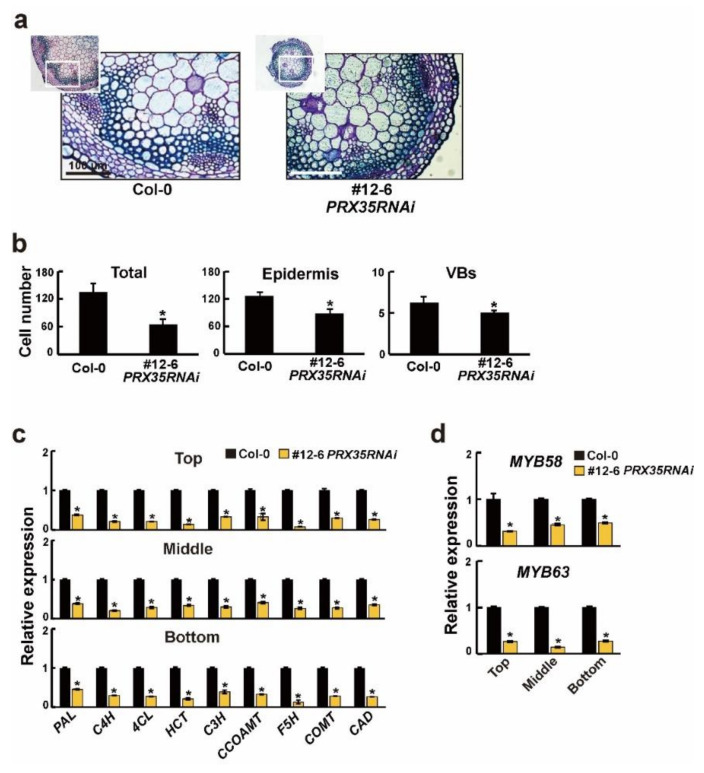
Stained cross-sections, cell numbers, and expression levels of lignin biosynthesis and biosynthesis-regulated genes in the main stems between wild-type (Col-0) and *PRX35RNAi* plants. (**a**) Toluidine blue stained cross-sections of paraffin-embedded basal stem portions in 6-week-old wild-type (Col-0) and *PRX35RNAi* plants. White boxes shown in upper left corner in each photo are magnified. Scale bars are presented in photos. (**b**) Total cell numbers of xylem cells and interfascicular fibers along the vascular ring, and the numbers of epidermal cells and vascular bundles (VBs). (**c**,**d**) Transcript levels of lignin synthesis (**c**) and lignin synthesis-regulated genes (**d**) in different regions of stems of wild-type (Col-0) and *PRX35RNAi* plants, measured by RT–qPCR. Expression levels in wild-type (Col-0) plants were defined as 1.0. Error bars indicate the standard error of the mean. The *UBQ10* gene was used as an internal control. Asterisks indicate significant differences (*p* < 0.005, Student’s *t*-test).

## Data Availability

All data are presented in the paper.

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
