# Peer review of "Reduced Expression of *PRX2*/*ATPRX1*, *PRX8*, *PRX35*, and *PRX73* Affects Cell Elongation, Vegetative Growth, and Vasculature Structures in *Arabidopsis thaliana"

_plants, 2022, doi:10.3390/plants11233353_

Round 1
Reviewer 1 Report
The manuscript by Jeong et al., entitled “Reduced expression of PRX2/ATPRX1, PRX8, PRX35, and PRX73 affects cell elongation, vegetative growth, and vasculature structures in Arabidopsis thalian”, selected four peroxidase genes (PRX2/ATPRX1, PRX8, PRX35, and PRX73) from a previous genome-wide transcriptome analysis (downregulated in phytochrome B (phyB)-77 mutants but were upregulated in brassinosteroids related bri1-5 mutants), and then created PRX2RNAi, PRX8RNAi, PRX35RNAi, and PRX73RNAi plants and performed phenotypic and morphological analyses. The phenotype is interesting and clear.
The study is quite straight forward and a simple story. The manuscript is well written. However, For the experimental design, some main concerns are listed below,
1) It’s good to see that two lines of each PRXRNAi plant were created for experimentation. One concern is why the author didn’t use any T-DNA mutants from Arabidopsis Biological Resource Center (ABRC: https://abrc.osu.edu/) or The European Arabidopsis Stock Centre (NASC: http://arabidopsis.info/)? Is it lethal or not available? If yes, the author can simply state it in the text.
2) Why the author didn’t create overexpression plants of these four PRXs genes to observe the phenotype? It’s much better to have it.
3) The discussion section can be improved. a) The author showed the phenotype of PRX2RNAi, PRX8RNAi, PRX35RNAi, and PRX73RNAi plants from Figure 2 to Figure 7, Whether are they showed some more special as compared to other known function of PRXs? b) In Figure 6C and 6D, it showed downregulation of lignin biosynthesis and biosynthesis-regulated genes in PRX35RNAi plants, this can be discussed a little more.
Other mirror comments,
1) The title “Arabidopsis thalian” to “Arabidopsis thaliana”.
2) Figure 1, figure legend, line 138. “b, c Relative expression levels of PRX genes in wild-type (Col-0) and respective PRXRNAi plants grown for 6 days. Relative expression levels measured by RT–qPCR in wild-type (Col-0) plants were defined as 1.0.” c should be explained separately and clearly. Take time to figure it out.
3) Figure 3, The flowering time looked quite earlier in mutants than WT, have you count rosette leaves number when the plants close to blotting? It can help to evaluate the earlier flowering time was caused by faster growth or changes of flowering time control genes. It can be discussed in the discussion section.
Author Response
Response to Reviewer 1:
Thank you very much for your kind reviews and comments. We believe your suggestions can improve our manuscript significantly. We have revised our manuscript according to your suggestions. We hope you can agree with these changes in the present version. All changes made to the text are shown in track change so that you can be easily identified. And the responses are shown as follows.
Q1. It’s good to see that two lines of each PRXRNAi plant were created for experimentation. One concern is why the author didn’t use any T-DNA mutants from Arabidopsis Biological Resource Center (ABRC: https://abrc.osu.edu/) or The European Arabidopsis Stock Centre (NASC: http://arabidopsis.info/)? Is it lethal or not available? If yes, the author can simply state it in the text.
Response: We agree with Reviewer1’s opinion. We have already a single T-DNA mutant, in which T-DNA insertions were found in the promoter, 5’-untranslated region (UTR), or 3’-UTR, for each four PRX genes and found no apparent phenotypic changes in their T-DNA mutants. However, because we just isolated and characterized only one single T-DNA mutant for each four PRX gene, we did not include their data in our manuscript. As suggested by Reviewer1, we included this data in Figure S3 of our revised manuscript.
Q2. Why the author didn’t create overexpression plants of these four PRXs genes to observe the phenotype? It’s much better to have it.
Response: We agree with Reviewer1’s opinion. Because 35S promoter-driven transgene(s) lead to ectopic or highly elevated expression in transgenic plants (we have already discussed this point in the Discussion part), we will make their own endogenous promoter-drive sense and antisense transgenic plants (we modified this point in the Discussion part), and characterize their phenotypes in the near future.
Q3. The discussion section can be improved. a) The author showed the phenotype of PRX2RNAi, PRX8RNAi, PRX35RNAi, and PRX73RNAi plants from Figure 2 to Figure 7, Whether are they showed some more special as compared to other known function of PRXs? b) In Figure 6C and 6D, it showed downregulation of lignin biosynthesis and biosynthesis-regulated genes in PRX35RNAi plants, this can be discussed a little more.
Response: We appreciate this insightful comment raised by Reviewer1. We included these points in the Discussion part of our revised manuscript.
Q4. The title “Arabidopsis thalian” to “Arabidopsis thaliana”.
Response: When we checked title’s typo in our submitted file, the title was right. We again confirmed it.
Q5. Figure 1, figure legend, line 138. “b, c Relative expression levels of PRX genes in wild-type (Col-0) and respective PRXRNAi plants grown for 6 days. Relative expression levels measured by RT–qPCR in wild-type (Col-0) plants were defined as 1.0.” c should be explained separately and clearly. Take time to figure it out.
Response: We included the expression levels of target genes in wild-type plants and statistical analyses in Figure 1c of our revised manuscript.
Q6. Figure 3, The flowering time looked quite earlier in mutants than WT, have you count rosette leaves number when the plants close to blotting? It can help to evaluate the earlier flowering time was caused by faster growth or changes of flowering time control genes. It can be discussed in the discussion section.
Response: We agree with Reviewer 1’s opinion that including data showing determination of rosette leaves number and the expression levels of flowering time genes in four PRXsRNAi lines will certainly provide clear evidence regarding the involvement of these peroxidases in the control of flowering time. At this stage, we focused on a variety of morphological changes such as root and hypocotyl elongation, vegetative growth, and the vascular structures in several tissues in our four PRXsRNAi plants. We will perform further experiments on flowering time in our four PRXsRNAi lines. We included this point in the Discussion part of our revised manuscript.

Reviewer 2 Report
The manuscript entitled “Reduced expression of PRX2/ATPRX1, PRX8, PRX35 and PRX73 affects cell elongation, vegetative growth, and vasculature structure in Arabidopsis thaliana” by Jeong et al. showed that four Class III peroxidases that are regulated under phytochrome and brassinosteroid signaling are involved in several processes in plant development such as seedling growth, flowering, leaf senescence, so on. To examine functions of PRXs in developmental processes, the authors generated knockdown plants of the four PRX genes by RNA interference (RNAi) and conducted phenotypic analyses of the plants. Class III peroxidases are one of the most common enzyme families in plants and are required for metabolism of reactive oxygen species (ROS) that play important roles as growth regulators and signaling molecules. Therefore, new findings of PRXs would be helpful in related-research fields. However, there are problems in the manuscript that need to be resolved.
1) The most concern in this study is that several conclusions of PRX35RNAi plants have been based on results from only one plant line (#12-6). It cannot be excluded a possibility that aberrant phenotypes of PRX35RNAi may be caused by effect of T-DNA insertion but not of RNAi. This is fundamental to discuss function of PRX35 in this manuscript. Please show results from independent transgenic lines.
2) Quality of most microscope images in Figures 4-7 are not enough for showing abnormalities that the authors found. Please provides images in proper resolution and magnification. Color information is also quite important to compare differences in senescent leaves and lignin deposition in dye-stained stem sections in Figures 3b (leaf color is different even in Col-0 among the images) and 6a, respectively. Please take photos under the same micrograph condition.
3) The authors say that they specifically designed each region of four PRX genes for RNA silencing in Line 121. However, according to Method section, they used full-length ORF sequences of each PRX gene as triggers of RNAi. Because of 75-80% sequence identities among the PRX genes shown by the authors, it is expected that specificity of those RNAi triggers for each target gene would be quite low as resulted in Figure 1c. In case that the description in Method section is correct, please remove “specifically” from Line 121 in order to avoid confusion and add explanation of the reason why full-length ORF was used for this study.
4) It seems that T-DNA insertion mutants of the four PRX genes are available from public Arabidopsis resource centers. It may be related to my comment above, why did the authors use RNAi plants for analysis of the PRXs but not T-DNA insertion mutants?
5) Figure 1c: Statistic analyses must be performed containing wild type but not only among RNAi plants so that non-specificity of the trigger sequences can be assessed in each plant.
6) In the first paragraph of Section 2.3, the authors have concluded that rosette leaves were morphologically changed to enlarge in RNAi plants, compared with wild type. However, leaves of several RNAi plants were significantly reduced their petiole length and leaf width or showed no significant changes. It is difficult to make conclusion if phenotypes are caused by effect of a transgene (RNAi), based on the results from only one transgenic plants in genetics study. Therefore, it is still unclear if PRX8 and PRX35 are involved in morphological changes of petiole length and leaf width. Please tone down this conclusion.
7) Figures 4a and 4b: It seems that cell size of RNAi plants is much larger than that of Col-0. Are the photos provided in the same magnification? If scale information is correct, it would be better to compare cell size among the tested plants in addition to cell number because it may give us information why RNAi plants grow larger than wild type.
8) In Figure 5, 5c and 5e is the same image.
9) The authors have conducted scanning electron microscopy for measuring cell wall thickness in stem tissues in Figure 7 and S6. As observed in the provided photos, tissues are largely shrieked. If they used such micrograph for measuring cell wall thickness, it would be difficult to compare it correctly.
Minor points are as follows:
10) Line 4: “thalian” should be “thaliana”.
11) Line 112: This sentence has been repeated just below (Line 120). So, it would be better to remove from Section 2.1.
12) Line 114: There is no period at the end of the sentence.
13) Line 153: This paragraph describes morphological phenotypes. Please move to Section 2.3.
14) Figure 2b: Based on the result shown in Figure 2a, roots in RNAi plants were longer than that of wild type. Please avoid to use minus value in Relative length because minus value means to be shorter than control.
15) Figure 3e and 3f: Unit information is not available in chlorophyll and anthocyanin contents in the graphs.
16) Line 198: “Fig. 4e” should be “Fig. 3e”.
17) Table S1: No information is available what values in the table. For helping readers understand, please add explanation.
Author Response
Response to Reviewer 2:
Thank you very much for your kind reviews and comments. We believe your suggestions can improve our manuscript significantly. We have revised our manuscript according to your suggestions. We hope you can agree with these changes in the present version. All changes made to the text are shown in track change so that you can be easily identified. And the responses are shown as follows.
Q1. The most concern in this study is that several conclusions of PRX35RNAi plants have been based on results from only one plant line (#12-6). It cannot be excluded a possibility that aberrant phenotypes of PRX35RNAi may be caused by effect of T-DNA insertion but not of RNAi. This is fundamental to discuss function of PRX35 in this manuscript. Please show results from independent transgenic lines.
Response: We have already included the data showing the changes of vascular structures in two independent PRX35RNAi lines (#12-6 and #19-8) in Figures 4, 5, and 7, and Figures S5, 6, and 7. However, the resolution of some photos was not good and so that we presented them in Supplementary Figures. As Reviewer2 pointed out, we included two independent PRX35RNAi line numbers (#12-6 or #19-8) in Section 2.5 and 2.6 of the Results part, and also moved the data of #19-8 PRX35RNAi line in Figure S7 to Figure 7. Furthermore, we described the possible function of PRX35 in the Discussion part of our revised manuscript.
Q2. Quality of most microscope images in Figures 4-7 are not enough for showing abnormalities that the authors found. Please provides images in proper resolution and magnification. Color information is also quite important to compare differences in senescent leaves and lignin deposition in dye-stained stem sections in Figures 3b (leaf color is different even in Col-0 among the images) and 6a, respectively. Please take photos under the same micrograph condition.
Response: As Reviewer2 pointed out, we changed some photos in Figures. For instance, we similarly adjusted the contrast of each photo according to wild-type leaves’ color in Figure 2b and also adjusted the color of cross-section in wild-type plants in Figure 6a. Furthermore, we changed Figure 5e to show the difference of cell wall thickness between wild-type and PRX35RNAi plants.
Q3. The authors say that they specifically designed each region of four PRX genes for RNA silencing in Line 121. However, according to Method section, they used full-length ORF sequences of each PRX gene as triggers of RNAi. Because of 75-80% sequence identities among the PRX genes shown by the authors, it is expected that specificity of those RNAi triggers for each target gene would be quite low as resulted in Figure 1c. In case that the description in Method section is correct, please remove “specifically” from Line 121 in order to avoid confusion and add explanation of the reason why full-length ORF was used for this study.
Response: As Reviewer3 pointed out, we deleted “specifically” and included the reason why we used full-length ORF region of four PRX genes for RNA silencing in the Introduction part of our revised manuscript.
Q4. It seems that T-DNA insertion mutants of the four PRX genes are available from public Arabidopsis resource centers. It may be related to my comment above, why did the authors use RNAi plants for analysis of the PRXs but not T-DNA insertion mutants?
Response: We agree with Reviewer2’s opinion. We have already a single T-DNA mutant, in which T-DNA insertions were found in the promoter, 5’-untranslated region (UTR), or 3’-UTR, for each four PRX genes and found no apparent phenotypic changes in their T-DNA mutants. However, because we just isolated and characterized only one single T-DNA mutant for each four PRX gene, we did not include their data in our manuscript. As suggested by Reviewer1 and 2, we included this data in Figure S3 of our revised manuscript.
Q5. Figure 1c: Statistic analyses must be performed containing wild type but not only among RNAi plants so that non-specificity of the trigger sequences can be assessed in each plant.
Response: We included the expression levels of target genes in wild-type plants and statistical analyses in Figure 1c of our revised manuscript.
Q6. In the first paragraph of Section 2.3, the authors have concluded that rosette leaves were morphologically changed to enlarge in RNAi plants, compared with wild type. However, leaves of several RNAi plants were significantly reduced their petiole length and leaf width or showed no significant changes. It is difficult to make conclusion if phenotypes are caused by effect of a transgene (RNAi), based on the results from only one transgenic plants in genetics study. Therefore, it is still unclear if PRX8 and PRX35 are involved in morphological changes of petiole length and leaf width. Please tone down this conclusion.
Response: We agree with Reviewer2’s opinion. We toned down this paragraph in the Results part of our revised manuscript.
Q7. Figures 4a and 4b: It seems that cell size of RNAi plants is much larger than that of Col-0. Are the photos provided in the same magnification? If scale information is correct, it would be better to compare cell size among the tested plants in addition to cell number because it may give us information why RNAi plants grow larger than wild type.
Response: We agree with Reviewer2’s opinion. Although we did not check quantified data of cell size between PRXRNAi lines and wild-type plants, we described this point in the Results part of our revised manuscript.
Q8. In Figure 5, 5c and 5e is the same image.
Response: We changed the figures in Figure 5e of our revised manuscript.
Q9. The authors have conducted scanning electron microscopy for measuring cell wall thickness in stem tissues in Figure 7 and S6. As observed in the provided photos, tissues are largely shrieked. If they used such micrograph for measuring cell wall thickness, it would be difficult to compare it correctly.
Response: We agree with Reviewer2’s opinion. Although some tissues shown in Figure 7 and S7 were shrieked during the sample preparation for SEM analyses, we measured cell wall thickness in total 30 cells (each 10 cell from three individual plants for PRX35RNAi lines) as described in Figure 7 legend. Thus, we tried to correctly obtain the quantified data.
Q10. Line 4: “thalian” should be “thaliana”.
Response: When we checked title’s typo in our submitted file, the title was right. We again confirmed it.
Q11. Line 112: This sentence has been repeated just below (Line 120). So, it would be better to remove from Section 2.1.
Response: As Reviewer3 pointed out, we modified the sentences in the Introduction part of our revised manuscript.
Q12. Line 114: There is no period at the end of the sentence.
Response: We corrected it.
Q13. Line 153: This paragraph describes morphological phenotypes. Please move to Section 2.3.
Response: As Reviewer3 pointed out, we moved this paragraph to Section 2.3 in the Results part of our revised manuscript.
Q14. Figure 2b: Based on the result shown in Figure 2a, roots in RNAi plants were longer than that of wild type. Please avoid to use minus value in Relative length because minus value means to be shorter than control.
Response: We modified the Figure 2b, as pointed out by Reviewer3.
Q15. Figure 3e and 3f: Unit information is not available in chlorophyll and anthocyanin contents in the graphs.
Response: We included unit information in Figure 3e and 3f.
Q16. Line 198: “Fig. 4e” should be “Fig. 3e”.
Response: We corrected it in Figure 3 legend.
Q17. Table S1: No information is available what values in the table. For helping readers understand, please add explanation.
Response: We modified Table S1.

Reviewer 3 Report
Title: Should be ‘Arabidopsis thaliana’
Introduction: I am not very convinced about the ‘functional redundance of class III peroxidases’ (lines 45 and 54). The reference the authors site is as old as from 2009. We know that the substrate specificities of the different enzymes may be similar, but as the corresponding genes are expressed in various tissues and different developmental states, and hence in real tissues they may have very specific functions. I would suggest reconsidering the word redundancy here in the light of present data (as in this manuscript, too!). Information between lines 62 and 77 also point to specific functions.
Line 83. I do not understand the ‘reduce the flame retardancy of biomass’ in this context. Are the authors meaning ‘recalcitrance of biomass caused by lignin’? Please explain.
Figure 4. This is a minor matter but for future I suggest the authors to always show the photomicrographs with tissies the same orientation (b), similarly as in the upper row (a). It makes it easier for the reader to compare the photomicrographs.
Lines 330-335, 369-370 and 401-404 all explain the reduction in lignified cell numbers and lignification levels in the RNAi lines as due to the reduced expression levels of lignin-biosynthesis and synthesis-regulated genes. Is it rather the other way: due to feedback inhibition caused by accumulation of monolignols that is due to reduced class III peroxidase levels? If so, please correct accordingly.
Author Response
Response to Reviewer 3:
Thank you very much for your kind reviews and comments. We believe your suggestions can improve our manuscript significantly. We have revised our manuscript according to your suggestions. We hope you can agree with these changes in the present version. All changes made to the text are shown in track change so that you can be easily identified. And the responses are shown as follows.
Q1. Title: Should be ‘Arabidopsis thaliana’
Response: When we checked title’s typo in our submitted file, the title was right. We again confirmed it.
Q2. Introduction: I am not very convinced about the ‘functional redundance of class III peroxidases’ (lines 45 and 54). The reference the authors site is as old as from 2009. We know that the substrate specificities of the different enzymes may be similar, but as the corresponding genes are expressed in various tissues and different developmental states, and hence in real tissues they may have very specific functions. I would suggest reconsidering the word redundancy here in the light of present data (as in this manuscript, too!). Information between lines 62 and 77 also point to specific functions.
Response: We appreciate this insightful comment raised by Reviewer3. We modified this paragraph in the Introduction part of our revised manuscript. We also updated some references.
Q3. Line 83. I do not understand the ‘reduce the flame retardancy of biomass’ in this context. Are the authors meaning ‘recalcitrance of biomass caused by lignin’? Please explain.
Response: As Reviewer3 pointed out, we modified it.
Q4. Figure 4. This is a minor matter but for future I suggest the authors to always show the photomicrographs with tissues the same orientation (b), similarly as in the upper row (a). It makes it easier for the reader to compare the photomicrographs.
Response: We appreciate this insightful comment. We added the label (IF and Pi) in Figure 4b to clarify significant changes in the stained cells between WT and four PRXRNAi plants.
Q5. Lines 330-335, 369-370 and 401-404 all explain the reduction in lignified cell numbers and lignification levels in the RNAi lines as due to the reduced expression levels of lignin-biosynthesis and synthesis-regulated genes. Is it rather the other way: due to feedback inhibition caused by accumulation of monolignols that is due to reduced class III peroxidase levels? If so, please correct accordingly.
Response: We appreciate this insightful comment. We modified this paragraph in the Results and Discussion parts of our revised manuscript.

Round 2
Reviewer 1 Report
My comments/concern were addressed well by the authors.
Author Response
Dear Reviewer1,
Thank you for your useful comments.
Reviewer 2 Report
The revised manuscript was somehow improved in text based on reviewers’ comments. It is good to be newly added results from T-DNA insertion mutants. But, it still remains to be addressed crucial problems.
1) I would not like to repeat my previous comment. But no significant improvement was found in microscope images of Figures, especially SEM, from the previous manuscript. It is quite difficult to find abnormalities pointed out by the authors. Also, manipulation of image contrast is worse way when compare differences in color between images.
2) Line 130: The authors concluded that T-DNA insertion mutants caused no apparent phenotypic changes. However, it looks like that prx35 showed early flowering and accelerated leaf senescence phenotypes (Fig S3b and S3c). Since the authors mentioned flowering and leaf senescence phenotypes of knockdown plants in this study, please observe and quantify phenotypes carefully.
3) Figure 2b: It is unclear what is targets (sample pairs) of statistical analysis shown by the asterisk.
4) Line 216: In the second paragraph of section 2.3, the authors concluded that the morphological parameters of rosette leaves in RNAi plants remained unaltered compared with those of wild type. However, several plants showed significant differences in such parameters like PRX2RNAi. Please make conclusion based on the results.
5) Line 279: Although there is description that two independent PRX35RNAi lines (#12-6 and #19-8) showed severe changes in the size of the main stem and the number of vascular bundles, no results have been shown about #19-8. The authors must provide quantified data to demonstrate such phenotypes in the both lines.
6) Line 317: Epidermal, cortical and endodermal cells are not vascular system cells. Therefore, the conclusion of the sentence “the root vascular system of PRX35RNAi plants had differentiated in a regular pattern” is strange.
7) Line 343: What does “lignin degree” mean? Maybe the author want to say “degree of polymerization”?
Minor points are as follows:
8) Line 306: “6b” should be “5b”.
9) Line 309: “6e” should be “5e”.
Author Response
Response to Reviewer 2:
Thank you very much for your kind reviews and comments. We believe your suggestions can improve our manuscript significantly. We have revised our 1st ㄱrevised manuscript according to your suggestions. We hope you can agree with these changes in the present version. All changes made to the text are shown in track change so that you can be easily identified. The responses are also shown as follows.
Q1. I would not like to repeat my previous comment. But no significant improvement was found in microscope images of Figures, especially SEM, from the previous manuscript. It is quite difficult to find abnormalities pointed out by the authors. Also, manipulation of image contrast is worse way when compare differences in color between images.
Response: We agree with Reviewer2’s opinion. Because our samples used in SEM analysis were not good and so they were not suitable for quantification of thickness of secondary cell walls in xylem tissues and interfascicular fibers between wild-type and PRXRNAi plants, and several data included in our manuscript showed the alterations in cell walls in vascular regions of PRXRNAi plants, we deleted these SEM data shown in Figure 7 and S7 in our 2nd revised manuscript.
Q2. Line 130: The authors concluded that T-DNA insertion mutants caused no apparent phenotypic changes. However, it looks like that prx35 showed early flowering and accelerated leaf senescence phenotypes (Fig S3b and S3c). Since the authors mentioned flowering and leaf senescence phenotypes of knockdown plants in this study, please observe and quantify phenotypes carefully.
Response: As Reviewer2 pointed out, we described a slightly early flowering and more accelerated leaf senescence phenotypes of a single prx35 mutants, and included its bolting days shown in Figure S3e in our 2nd revised manuscript.
Q3. Figure 2b: It is unclear what is targets (sample pairs) of statistical analysis shown by the asterisk.
Response: As Reviewer2 pointed out, we modified Fig. 2b in our 2nd revised manuscript.
Q4. Line 216: In the second paragraph of section 2.3, the authors concluded that the morphological parameters of rosette leaves in RNAi plants remained unaltered compared with those of wild type. However, several plants showed significant differences in such parameters like PRX2RNAi. Please make conclusion based on the results.
Response: As Reviewer2 suggested, we rephrased it in our revised manuscript as follows: “The morphological parameters (petiole length, blade length, and blade width) of rosette leaves in the PRX2RNAi and PRX73RNAi plants were significantly altered compared with those of the wild-type (Col-0) plants, whereas leaves of some PRX8RNAi and PRX35RNAi lines showed reduced or no significant changes in their petiole length and leaf width” in our 2nd revised manuscript.
Q5. Line 279: Although there is description that two independent PRX35RNAi lines (#12-6 and #19-8) showed severe changes in the size of the main stem and the number of vascular bundles, no results have been shown about #19-8. The authors must provide quantified data to demonstrate such phenotypes in the both lines.
Response: As Reviewer2 pointed out, we included the quantified data in Fig. S5d of our 2nd revised manuscript.
Q6. Line 317: Epidermal, cortical and endodermal cells are not vascular system cells. Therefore, the conclusion of the sentence “the root vascular system of PRX35RNAi plants had differentiated in a regular pattern” is strange.
Response: As Reviewer2 pointed out, we deleted in in our 2nd revised manuscript.
Q7. Line 343: What does “lignin degree” mean? Maybe the author want to say “degree of polymerization”?
Response: Because it was difficult to compare the degree of polymerization of lignin between wild-type and PRX35RNAi plants from phloroglucinol-HCl staining data shown in Figure 6a of our 1st revised manuscript, we decided to delete this data in our 2nd revised manuscript. We also rephrased the sentences in the paragraph of section 2.6.
Q8. Line 306: “6b” should be “5b”.
Response: We corrected it.
Q9. Line 309: “6e” should be “5e”.
Response: We corrected it.
